# Research on Evaluation Indexes and Weights of the Aging-Friendly Community Public Environment under the Community Home-based Pension Model

**DOI:** 10.3390/ijerph17082863

**Published:** 2020-04-21

**Authors:** Wen-Bing Mei, Che-Yu Hsu, Sheng-Jung Ou

**Affiliations:** 1Department of Art Design, Guangdong Industry Polytechnic, Guangzhou 510300, China; mlkun3101@gmail.com; 2Department of Architecture, Chaoyang University of Technology, Taichuang 41349, Taiwan; cyhsu428@cyut.edu.tw

**Keywords:** community public environment, aging-friendly, FAHP

## Abstract

Community home-based care has become China’s main mode of care for the elderly, and the aging of the community public environment has become the focus of attention of all of society. This study uses a questionnaire survey and the fuzzy analytic hierarchy process (FAHP) to (i) obtain the relative weights of indicators in the hierarchy structure of an aging-friendly community public environment and (ii) build a complete indicator evaluation system for the aging-friendly community public environment. The research results show that the quasi-side evaluation index framework of the aging-friendly community public environment is composed of four factors (i.e., community facilities, community road system, community environmental function, and community landscape configuration) and 24 evaluation indexes. The weights of the indicators in descending order are “community road system (w = 0.374)”, “community facilities (w = 0.310)”, “community environmental functions (w = 0.264)”, and “community landscape configuration (w = 0.052)”. The research results show that "community road systems” and “community facilities” are important indicators of the aging-friendliness of a community public environment. “Community environmental function” is an important supplemental factor of the aging-friendliness of a community public environment. “Community landscape configuration” involves improving the construction of the community public environment from the perspective of landscaping. Among all indicator levels, the weights of “Community road floor slip resistance” (w = 0.1795), “Daily health and medical facilities (w = 0.1181)”, and “Provide social interaction functions (w = 0.1067)” are ranked the highest. These results show that ensuring the physical and mental health of the elderly in the community is a core criterion for evaluating the aging-friendliness of a public environment in the community. In this study, an index evaluation weight system is established to clarify the best approach to constructing an aging-friendly community public environment in accordance with previous standard specifications. This system can further clarify the scientific method for evaluating aging-friendly public environments built in the past and can serve as a reference for the practical world.

## 1. Introduction

### 1.1. Research Background and Motivation

The aging of the population has been accompanied by changes in family structure and functions caused by economic and social changes, and as a result, the family pension function has shifted to socialization. Faced with the steadily increasing demand for old-age care services brought about by aging, both in terms of the system and actual resource allocation, China’s response appears to be relatively lagging [1]. In the traditional Chinese family endowment mode, the elderly live with their children, and the children take care of the elderly. Because of the influence of modern factors such as urbanization, a declining birth rate, and globalization, it is becoming increasingly common for the elderly to live separately from their children. Especially under the dual influence of the declining birth rate and prolonged average life expectancy, China’s traditional family pension function is continuously weakening, and social pension pressure is gradually increasing. As a result of the shortage of old-age facilities, the high cost of old-age care, and the inadequate tenderness of old-age care, there is a structural contradiction between, on the one hand, a large number of elderly people and, on the other, long-term vacancies in professional old-age institutions. Therefore, the question of how to build an old-age care model that is in line with China’s national conditions and Chinese old-age care habits has become the focus of attention [2]. With economic development and the rise of the real estate industry in recent decades, the transition of residential buildings in China from original multi-story and small high-rise buildings to high-rise and super-high-rise buildings not only improves the living conditions of Chinese people, but also beautifies the community environment and sets up supporting facilities for the community. The functions of Chinese communities have been increasingly perfected, and the home-based care model has gradually emerged and developed into China’s most important care model. Community home-based care is able to not only improve the limitations of single and non-professional home care services but also overcome shortcomings, such as the high-end care costs and insufficient warmth of professional institutions. Relying on the community as a carrier, integrating various resources in the community, combining family pensions and institutional pensions, continuously enriching the community’s home pension facilities, and improving the life and quality of life of the elderly in the community, can lead to a creative transformation of the traditional pension culture and the perfection of China’s smart new pension model [3].

With the development of the economy and the improvement of material conditions, people’s livelihood demands must be transformed from the needs of material life to the needs of beautifying life. Material and cultural needs, such as old-age services, health services, and cultural activities, must also be met through economic behaviors, such as facility construction and functional improvement. To further increase the livelihood and welfare of the elderly, the current situation for the elderly in terms of the community residence, pensions, health care, daily life, social interaction, cultural entertainment, etc., must be made suitable for old-age habitability, through a combination of public facilities in the community and the requirements of the elderly. In 2016, the government authority proposed that the community should focus on building a suitable living environment, suitable health support environment, and a suitable living service environment. By 2025, a safe, convenient, and comfortable system for an aging-friendly community environment should be basically established [4]. It can be seen that under the community home-based care model, the aging community environment on which elderly people depend is particularly important. Building an aging-friendly community environment is important for ensuring the quality of life of the elderly and helping them to successfully age in the area. Therefore, elderly care urgently needs an environment for the elderly living community.

The problem of aging community construction in our country’s aging society has aroused widespread concern, and theoretical research and practical exploration have begun. However, at present, regardless of whether old communities are being renovated or new communities are being constructed, the aging-related conditions of the community public environment are mainly based on national laws, regulations, and standards. No community public environment has been built yet with the help of quantifiable aging assessment tools. 

On the basis of the above questions, this paper first reviews relevant national community home care regulations and domestic and foreign literature. Then, a questionnaire survey is used to initially construct an index evaluation system for the aging-friendly community public environment. On this basis, the fuzzy analytic hierarchy process (FAHP) is used to integrate the opinions of various experts on the impact indicators needed to evaluate the aging-friendliness of a community public environment and to assign weights to the relative importance of these indicators. On the basis of systematic reasoning, verification, and analysis, an attempt is made to establish a complete and rigorous index evaluation weight system for an aging-friendly community public environment, thereby providing research and practical references that can be used to evaluate the effectiveness of aging-friendly community public environment construction.

### 1.2. Research Purpose

In this study, professionally considered assessment criteria for aging-friendly community public environments were obtained by means of questionnaire surveys, to construct an assessment system for such environments. Through the fuzzy analytic hierarchy process (FAHP), the weights of the evaluation index of the aging-friendly community public environment were calculated. In sum, the purpose of this research can be described as follows:A questionnaire survey was used to first discover the key factors that affect the construction of an aging-friendly community public environment and then to build a hierarchy of evaluation indicators.The fuzzy analytic hierarchy process (FAHP) was used to explore the relative weights and rankings of the various indicators and to construct a complete community evaluation index system.The presented weighting system for the index evaluation of the aging-friendly community public environment can be used as a reference to evaluate the moderate aging of the community public environment, with the aim of providing practical solutions and research results for the cutting-edge planning of community public environment construction. It is expected that a suitable aging performance will be found, that meets the needs of the elderly in the community home.

## 2. Literature Review

This section provides the relevant research and discussions from the three aspects of the community home care model: the aging-friendly community public environment and the function and connotation of evaluation indicators of the aging-friendly community public environment. 

### 2.1. Community Home Care Model’s Connotation

Community home-based care is based on the community, combining family care and social care, that is, "home care in the community". Community home care introduces the characteristics of home care in the community to the characteristics of the nursing needs of the elderly in China, and it has become a new type of care model [5]. The concept of community home care was originally derived from “community care” in the UK. In the 1960s, Britain began to develop community care. After more than 20 years of development, it has become mature and increasingly perfected, and it has been recognized and emulated by various countries [6]. The term "community home care" was first proposed in the "Building a Community Home Care System", issued by China in 2001, and it generated active discussion and extensive practices in industry and academia. Community home-based care is formed on the basis of the existing home-based care model in China. It emphasizes the integration of the community into the operation of the care-based model. The "home-based care" in “community home-based care” comes from traditional family care. In 1995, Yuan Jihui introduced the concept of "home-based care" for the first time in the article "Strengthening the role of the family to support home support". With the aging of China and the evolution of traditional family pensions, home-based pensions started to receive widespread attention, as an effective way to improve and guarantee the quality of life of the elderly. From the current research perspective, the concept of "home-based care" mainly includes three aspects:Field theory: Home-based care allows the elderly to choose their own home as their place of care, instead of going to a special institution that concentrates on providing care for the elderly. This is the inheritance and development of the traditional way of supporting the elderly in the family, and it is a method that is in line with the wishes of the elderly in China. Home care is the optimization of family care, and it reflects the difference between the fields of home care and professional care [7].Auxiliary theory: Home-based care is not a kind of pension model that is tied to family care and institutional care; home-based care is an old-age care model that focuses on family care and is supplemented by social professional care services [8].Combination theory: Home-based care is essentially a compound care model. It is an organic combination of the family pension and institutional pension. On the one hand, the home-based care model strengthens the positive effects of environmental, material, and spiritual factors on family care for the elderly. On the other hand, the professional services, care, and medical care of social pension institutions are introduced to pension services to increase the function of home pensions [9].

The “community” in “community home-based care” for the elderly highlights the role of community integration. That is, the resources of social professional institutions can be provided to the elderly at home through the community platform. Communities can both deploy and integrate the functions of traditional family pensions and social institution pensions, and they can also serve as a platform for social pension resources to interact with grassroots governments, communities, and families to provide more convenient and diverse services to the elderly [10]. The community home care model emphasizes the platform and integration function of the community, because the community itself cannot provide sufficient resource support for home care; instead, it introduces social professional care agency resources through the community to provide more professional care services for home care. 

In the 1980s, as a result of dissatisfaction with the living atmosphere, living conditions, and the service model of the community-based professional institution’s pension model, Western society began to explore a set of elderly care models that were more humane and more in line with the needs of the elderly. These new models combined the psychological and physical characteristics of the elderly; thus, the concept of "aging in place" emerged [11]. "Aging in place" means that seniors retire in the homes and communities in which they have lived much of their lives. Aging in place can enable older people to spend their old age independently in a familiar environment, where they live independently, safely, and comfortably in the community in which they have lived for a long time. The elderly maintain their original habits and social relationships in a familiar community, which prevents feelings of loneliness that result from being away from a familiar community environment in the process of aging [12]. "Aging in place" is the closest to the functional meaning and geographical definition of community-based home care. The purpose of community-based home care is to enable the elderly to realize local care. On the one hand, community-based home care provides elderly people with convenient community life and diversified care services through community-based care facilities. On the other hand, in the process of interaction with the community, elderly people have a sense of belonging to the community, which is conducive to maintaining harmonious and stable mental health.

### 2.2. Connotation of the Aging-Friendly Community Public Environment

The aging community public environment refers to the sum of the various living environments and lives of the entire age group that encompasses the elderly. The public environment, in a narrow sense, refers to the physical environment of an entity; in a broad sense, it also includes the comprehensive environment of the economy, society, and culture. Ensuring the aging-friendliness of the community public environment is essential for improving the health conditions of the elderly, improving the quality of life of the elderly, and increasing safety assurances and opportunities for social participation, and it is thus a necessary basic condition for the elderly in the community [13]. Constructing an aging-friendly community public environment is the adjustment and creation of living space for the purpose of helping the elderly continue to live in a familiar environment. The American Elderly Act proposes that the definition of aging-friendly community reform is as follows: in order to meet the needs of people with physical and mental disabilities, through the implementation of adjustments and improvements in living space, the greatest possible elimination of space obstacles is to help the elderly to extend and maintain an independent and safe life as much as possible [14]. Therefore, the public environment of the aging community discussed in this thesis includes the two important concepts of "care facilities" and "accessibility" in terms of space conditions.

"Elderly care facility building design code" (GB50867-2013) defines "old care facility" as an elderly care facility that provides special and comprehensive services to the elderly from aspects of housing, living care, medical care, rehabilitation care, and spiritual comfort. A community care facility is an important carrier and support system for community home care. Because of the lack of relevant standards and classification, coupled with the multiple attributes and functions of nursing facilities, a unified professional classification method for community nursing facilities has not yet been formed. As a combination of the concept of the organic integration of the community environment and old-age services, community old-age service facilities should be integrated into community life, have the comfort and affinity of home, and fully interact with residents. At the same time, the community should minimize the facility atmosphere and create a living place instead of a sheltered place so that the elderly can maintain their personal autonomy [15]. The community care facilities discussed in this thesis are categorized as community home care facilities, which are types of facilities that combine multifunctional, miniaturized, networked community care for elderly people living in a community with self-care, assistance, and care needs facilities [16].

Although healthy elderly people do not have diseases, they still have inflexible legs and feet, decreased sensory function, decreased vision, hearing loss, and memory deterioration due to age [17]. Therefore, community accessibility is a response to the special requirements of elderly people living in the community. Accessibility means that facilities such as access roads, buildings, transportation, and communication services are in place for the autonomous, safe, equal, and convenient travel and participation of special groups, such as the elderly and the disabled [18]. Obstacles are factors that reduce a person’s capability or accessibility in the physical environment. The accessibility conditions of the community discussed in this thesis include not only conditions that meet the needs of the elderly’s daily behavior, but also the related conditions that meet the needs of the social interaction behavior of the elderly. Specifically, the barrier-free conditions of the public environment in the daily behavior of the elderly include the safety design and treatment of the ground, stairwells, ramps, outdoor lighting, etc., especially the application of anti-skid materials on the ground, additional ramps, well-placed handrails, and barrier-free landscapes, etc. Barrier-free conditions for seniors’ social communication include clear and easily recognizable safety instructions for seniors with visual or hearing impairment [19]. These facilities should also include ventilation, sunshine, quiet protection for elderly people’s long-term stays, rest spaces in pedestrian walkways to prevent elderly people from walking for a long time, emergency help facilities, and related cultural service facilities, living service facilities, and information-barrier-free facilities. To realize a community public environment without barriers, it is necessary to respect and consider the use requirements of the elderly and to reflect the concern and response to the elderly in the community on the basis of the principle of "normalization and generalization".

### 2.3. Function and Connotation of the Aging-Friendly Community Public Environment Evaluation Index

The successful construction of an aging-friendly community public environment depends on whether the aging-related construction standards can be implemented strictly and objectively afterward. This is because the construction of a community public environment must reflect the community’s vision for an aging-friendly community, and the government needs a set of guiding principles or standards to assess whether the construction of an aging-friendly community public environment can successfully achieve this vision. Thus, a complete evaluation system is needed.

An indicator is a measurement tool and refers to “more than one symbolic representation of an input, course, or result, established for different purposes. Indicators can be applied to comparisons over time, or as absolute standard comparisons, or between and within groups.” [20]. A good indicator system should have the following important characteristics:Theoretical: The indicators are supported by theoretical evidence.Hierarchy: The indicator system covers all areas and levels and can be processed separately, with a hierarchical relationship.Diversity: Multiple sources of information are used to show complex phenomena.Completeness: The indicators have a complete logical system [21].

From the above, it can be inferred that an ideal evaluation index of the aging-friendly community public environment can fully reflect the characteristics and rationality of community public environment construction and can meet various needs, such as social goals, government vision, user groups, community attributes, etc. The “indicator” referenced in this research has the meaning of evaluation. Evaluation refers to “determining the degree or value of things according to appropriate standards”, that is, to check the overall effectiveness of construction through indicators and detailed criteria. It should have an “effectiveness assessment”, “improvement basis and methods”, “characteristic development”, and “process to determine the degree of achievement of goals”, and it should “determine the degree or value of things based on appropriate standards” features [22,23,24]. The main purpose of this research is to develop an evaluation index system for the aging-friendly community public environment through rigorous procedures and further evaluate the effectiveness of its construction. By measuring the indicators, the aging-related construction standards of the community public environment and the development trend of the needs of the elderly can be observed. The basic components of indicators should include the dual meanings of "target" and "data" [25]. An evaluation index of aging-friendliness can be used to observe the overall phenomena of current aging environment and construction trends in the community public environment. It has an important influence on the development of the pension industry. Evaluation indicators must be related to measures of people’s quality of life, including economic, social, environmental, natural, and cultural factors [26]. In summary, an appropriate evaluation index of the community public environment should respond to the development of the pension industry in its new form. The construction of the community public environment strives to achieve standardization that is suitable for the elderly and highlights the uniqueness of the community environment. Therefore, assessment indicators of the aging-friendly community public environment need to be able to reflect, in a quantifiable way, the standardization of environmental construction, the needs of users, and the characteristics of the community, and it should reflect the users’ preference characteristics and the core value of community pension standards.

## 3. Materials and Methods 

The purpose of this study was to establish an aging-friendly community public environment indicator evaluation system and its weight values. The questionnaire survey method and fuzzy analytic hierarchy process (FAHP) were used to build an aging-friendly community public environment evaluation index system and calculate the weights of the evaluation indexes. The study was conducted in two parts. The aim of the first part was (i) to use a questionnaire to determine the most suitable factors for evaluating the construction of the aging-friendly community public environment and (ii) to use the factor analysis method to construct a hierarchy of evaluation indicators. The second part included a fuzzy-level expert questionnaire survey, which was used to quantitatively score the relationship between the evaluation indicators that affect the community public environment and aging; then, the decision analysis software Super Decisions was used to calculate the final weight values. The study period was from February 2019 to February 2020.

### 3.1. Part I: Constructing a Hierarchical Framework for the Evaluation of Community Public Environment Aging

The research in this section was carried out in three phases. In the first stage, relevant evaluation policies, laws, and regulations issued by the government were analyzed to collect potential evaluation indicators for the aging-friendliness of the community public environment; then, experts were invited to screen the indicators for fitness. The second stage was the administration of a questionnaire survey. The questionnaire targeted community-related elderly caregivers through the method of "WeChat questionnaire star", and factors with low reliability were eliminated. In the third stage, factor analysis was utilized to perform factor extraction and factor recombination, to build a hierarchy structure to evaluate the aging-friendly community public environment.

#### 3.1.1. Collection of Assessment Indicators for Community Public Environment Aging-Friendliness

The successful construction of an aging-friendly community public environment depends on the builder’s mastery of relevant pension policies, laws, and regulations. Its main purpose is to combine the pleasure and familiarity of the users’ community with basic functional needs, as well as meet the spiritual expectations of relevant people and pay attention to differences and characteristics of the community environment. On the basis of a semantic analysis of the pension policy and a keyword query in the literature, a total of 26 evaluation items were collected for this study. The main references used in the indicator collection are (1) “Opinions on Accelerating the Socialization of Social Welfare” issued by the Ministry of Civil Affairs in 2000, (2) “National Community Elderly Welfare Service Starlight Plan” issued by the Ministry of Civil Affairs in 2001, (3) “Notice on Carrying Out Socialized Demonstration Activities of Pension Services” issued by the Ministry of Construction in 2005, (4) “Standardization Construction Plan for the Elderly Service Industry (2013–2017)”issued by the State Council in 2011, (5) “Notice on Strengthening the Barrier-free Reconstruction of Public Facilities of Elderly Families and Residential Areas” issued by the Ministry of Construction in 2014, (6) “Responses to the Proposal on Establishing a Good Living Environment for the Elderly” issued by the Ministry of Civil Affairs in 2016, (7) “Several Opinions on Accelerating the Development of the Elderly Care Service Industry” issued by the State Council in 2018, (8) “Planning Strategies for Elderly Friendly Communities in Old and Aged Cities” [27], and (9) “Evaluation index system of community living well-being under home care model” [1]. Next, a team of experts was invited to conduct the first stage of index fitness screening and initially adjust and modify unsuitable indicators. The members of the expert group have rich practical experience, and the team leader is a professor in the field of design as a doctoral supervisor and has experience in design practice, design management, and design research. After the first stage of expert discussions, 26 evaluation items collected from the institute were used as the basis for the subsequent questionnaire survey.

#### 3.1.2. Questionnaire Survey and Index Selection of Community Public Environment Aging-Friendliness

The content of the questionnaire was divided into two parts: basic information and content items. Basic information included gender, age, education, occupation, monthly income, and the community building age. The content items were based on the importance and performance of the 26 indicators obtained from the foregoing research results, and the importance of each factor and the overall performance of the community public environment were tested. According to the preliminary evaluation index of experts, Likert 5 scale was designed to conduct questionnaire survey on the importance and performance for each index, where 1 is very unimportant (very dissatisfied); 2 is not important (dissatisfied); 3 is ordinary; 4 represents important (satisfied); 5 is very important (very satisfied). The fields involved in community home care for the elderly are quite diversified. This survey used the “WeChat Questionnaire Star” method to send the electronic questionnaire to relevant industry experts, community elderly, nursing staff, and elderly family members. A wide range of college student resources were used to help less educated or older test subjects to complete the questionnaire surveys to ensure the comprehensiveness of the tested samples. The questionnaire survey lasted for 30 days. A total of 205 questionnaires were sent, and the final number of samples received was 202. To reach “The number of questionnaire samples using factor analysis should be more than 5 times the number of variables in questionnaire.” [28]. In this study, the opinions of community-based elderly care providers were used as the source of research data. In order to ensure the reliability and validity of all measurement data, statistical data such as the mean (M) and standard deviation (SD) were used as the basis for judgment using SPSS statistical software (SPSS Inc., Chicago, IL, USA). The criteria for item deletion or modification principle were as follows: When “M ≥ 5.0”, the original item was maintained; when “5.0 > M > 4.5”, the expert was consulted as to whether it should be deleted or modified; when “M ≤ 4.5” and “SD ≥ 1”, the item was deleted. Then, the remaining indicators were screened through the alpha coefficient check method of project analysis. Usually, the Cronbach α coefficient value is a means of evaluating the internal consistency of items under the same factor. The researchers checked whether it met the requirements of a general reliability test, i.e., an α coefficient above 0.7 [29].

#### 3.1.3. Construction of a Hierarchical Framework for the Evaluation of Community Public Environment Aging-Friendliness

After processing the data from the questionnaire, factor extraction, factor recombination, and factor naming were performed through the factor analysis method in SPSS software. In addition to establishing a complete and scientific evaluation index of the key aspects of the aging-friendly community public environment, the hierarchy structure of the aging-friendly community public environment assessment was constructed.

### 3.2. Part II: Establishing Relative Weighting Values for the Evaluation Index of Community Public Environment Aging-Friendliness

#### 3.2.1. Questionnaire Survey of Fuzzy Hierarchy Analysis Experts

The aim of this stage was mainly to understand the relationship between the evaluation indicators of the community public environment and aging. The weight of the evaluation index of the aging-friendly community public environment was calculated by using a fuzzy hierarchy analysis expert questionnaire. With this evaluation method, a relative weight ratio was used, and "relatively important values", the "acceptable maximum value", and the "acceptable minimum value" were selected. The collected data were sorted, compared, and checked according to the relative importance of the indicators. The ratio was subjectively determined on the basis of the professional and academic experience of the experts. The instructions for filling out the fuzzy level analysis expert questionnaire are listed in Table 1 and Table 2.

#### 3.2.2. Expert Selection and Questionnaire Distribution

The scope of the assessment system for the aging-friendly community public environment is broad and complex. If only one type of expert commented on the issue of aging, one would suspect that the information obtained was too biased and unreliable. According to Meltsner (1976), the technical indicators for selecting experts are "politics" and "analysis" [30]. In this research, experts and scholars with relevant research, teaching, and practical experience in environmental planning, social security, industrial design, etc., were invited to put forward valuable opinions during the expert investigation stage of this research, in order to achieve an appropriate weight calculation to assess the aging-friendliness of the community public environment. The survey began on November 10, 2019, and the deadline for the survey was December 30, 2019. A total of 12 valid questionnaires were received. The statistics of the expert datasheet is described in Table 3.

#### 3.2.3. Statistical Analysis of Questionnaire Data and Calculation of Indicator Weights

After the expert questionnaires were collected, a statistical analysis of the data was performed immediately, and weight analysis was performed for each level of the indicators. The fuzzy analytic hierarchy process (FAHP) in the system engineering method is more suitable for the scientific calculation of index weights in the evaluation system. The fuzzy analytic hierarchy process (FAHP), first proposed by Saaty, is a multi-objective decision-making method that combines quantitative and qualitative analysis [31]. FAHP improves the traditional analytic hierarchy process by addressing specific issues (such as a non-scientific basis for the difference between judgment consistency and matrix consistency) and improving the reliability of decisions. For a quantitative evaluation index, the selection of the optimal scheme provides a basis that has been widely used in the fields of society and management [32]. The operation mode is based on the research of Satty (1994) [33]. Pairwise comparisons between indicators were judged by experts. The evaluation scale is divided from "strong" to "weak", from "9:1" to "1:9", into 17 measures, and the relative importance of the indicators is assessed by marking the best value, maximum acceptable value, and minimum acceptable value. When using FAHP to make group decisions, Saaty recommends using geometric means to obtain group opinions. Therefore, when there are n experts, their scores are ×1, ×2, ×3,..., × *n*. The geometric mean calculation formula is as follows:(1)$$x=\sqrt[{n}]{x1*x2*x3*\ldots *xn}$$

After the above process, the geometric mean data from the experts were input into Super Decisions statistical software for a weight analysis and a consistency check of each level and index. Finally, calculations were performed to obtain the weight values and ranking of the “Aging-friendly Community Public Environment Evaluation Index”.

## 4. Results

Through questionnaire surveys and data analysis, the following results were obtained in this study.

### 4.1. Construction of the Hierarchical Framework of the Evaluation Indexes for the Aging-Friendly Community Public Environment 

#### 4.1.1. Screening and Interpretation of the Evaluation Indicators of the Aging-Friendly Community Public Environment 

A total of 26 evaluation indicators were selected through a review of the laws, policies, regulations, and literature related to the pension industry, as well as a preliminary evaluation by a team of experts. The interpretation of the 26 indicators for questionnaires and the evaluation indicators are shown in Table 4. The targeted questionnaire survey was administered via "WeChat Questionnaire Star". After the questionnaire data were collected, the average (M) and standard deviation (SD) of the 26 evaluation indicators were used as the basis for the evaluation. The average value of the 26 indicators of the statistical results was “M ≥ 4.5 and SD ≤ 1”, which met the requirements; therefore, 26 items/indicators were maintained for the subsequent parts of this study.

#### 4.1.2. Result of Factor Analysis 

In this study, the principal component analysis of factor analysis was used to extract and name the factors of the aging-friendly community public environment assessment. After rotation, normalization factor loads below 0.55 were deleted and regarded as items that are not ideal for naming [34]. The deleted items were "Community noise" and "Community lighting". In the process of factor analysis, principal component and Varimax rotation were used to extract the evaluation factors. The eigenvalue of factor 1 was 5.429, the eigenvalue of factor 2 was 4.213, the eigenvalue of factor 3 was 3.912, and the eigenvalue of factor 4 was 3.750, which explained 50.725%, 8.816%, 6.334%, and 6.226% of the variation in the variables, respectively. Thus, four factors explained 72.101% of the total variation. The Cronbach’s α of each factor and the total Cronbach’s α value were also high, which shows that the reliability of the study data was quite good (Table 5).

#### 4.1.3. The Naming and Hierarchical Structure of the Evaluation Index of the Aging-Friendly Community Public Environment 

The factor 1 evaluation indicators are “Public health facilities”, “Daily health care”, “Emergency relief facilities”, “Community public lighting”, “Shopping facilities”, “Signage facilities”, and “Provision rest seats”. Factor 1 is related to "facilities" in the public environment of the community, so it is named "facilities". The factor 2 evaluation indicators are “Community road anti-skid”, “Community road lighting”, “Accessibility of roads”, “The pedestrian path”, and ‘‘Road orientation”. Factor 2 is related to community roads, so it is named "road system". The factor 3 evaluation indicators are “Science education”, “Social communication”, “Cultural entertainment”, “Fitness function”, “Road open space”, and “Community ventilation”. Factor 3 is related to the functions suitable for the elderly provided by the community public environment, so it is named "environmental functions". The factor 4 evaluation indicators are “Road landscape”, “Ground landscape”, “Public landscape”, “Landscape accessibility”, “Plant health”, and “Water sculpture”. Factor 4 is related to the landscape configuration of the community public environment and community greening, so it is named “landscape greening”. On the basis of the above results, an evaluation index system was constructed for the community public environmental aging under the community home care model (Figure 1).

### 4.2. Calculation of Weights of Evaluation Indicators of the Aging-Friendly Community Public Environment 

One of the purposes of this study is to establish the relative weights of the various aspects and indicators in the hierarchy structure of the evaluation index hierarchy of the aging-friendly community public environment. The results of the expert evaluation of indicators and the statistical data from the analysis using “Super Decisions” are in Table 6.

## 5. Discussion

### 5.1. Analysis and Discussion of Relative Weight Values of Overall Indicators

The consistency ratio test for the aging-friendliness evaluation indicators in the four factors of the community public environment is 0.022 < 0.1. According to Saaty this consistency is acceptable [33]. Comparing the overall relative weights of the four factors, the highest is “road system” (W = 0.374), the second is “facility” (W = 0.310), the third is “environmental function” (W = 0.264), and the fourth is “landscape greening” (W = 0.052). From the statistical results of the relative weight order of the above four factors, the experts determined that “road system’’ and “facilities” are the two core criteria for the aging of the public environment, while "environmental function” is an effective supplement to the aging of the public environment, and “landscape greening” is more to improve the construction of the community public environment from the perspective of landscaping [35].

For the relative weight statistics of the 24 evaluation indicators, the consistency ratio test result is 0.033 < 0.1. According to Saaty this consistency is acceptable [33]. Out of all 24 evaluation indicators, the top three are "Community road anti-skid" (w = 0.1795), "Daily health care" (w = 0.1181), and "Social communication" (w = 0.1067). The research shows that the core elements of the community public environment are the assurance of the safety of elderly people living in the community and the provision of facilities and functions that meet their physical and mental health needs [36].

### 5.2. Analysis and Discussion of the Relative Weights of Detailed Evaluation Indicators

Among the seven evaluation indicators of "facilities", the order of importance is ‘‘Daily health care” (w = 0.381) > ”Public health facilities” (w = 0.143) > ”Emergency relief facilities” (w = 0.142) > ”Shopping facilities” (w = 0.141) > ‘‘Community public lighting” (w = 0.078) > ‘‘Provision rest seats” (w = 0.71) > ”Signage facilities” (w = 0.44). The weight of "Daily health care" is significantly higher than that of the other indicators. The results show that because of the decline in physical function, elderly people living in the community at home are in need of daily health care. It can be seen that daily health and medical facilities are an important basis for measuring whether the community facilities are aging-friendly.

Among the five evaluation indicators of the "road system", the order of importance is ‘‘Community road anti-skid” (w = 0.480) > ‘‘Accessibility of roads” (w = 0.220) > ‘‘Community road lighting” (w = 0.142) > ‘‘The pedestrian path” (w = 0.113) > ”Road orientation” (w = 0.045). The results show that for elderly people living in communities at home, safety (Community road anti-skid) and accessibility (accessibility of roads) are the most important evaluation indicators of community aging-friendliness, and security is significantly more important than accessibility.

Among the six evaluation indicators of the "environmental functions", the order of importance is “Social communication’’ (w = 0.404) > ‘‘Fitness function” (w = 0.228) > ‘‘Cultural entertainment” (w = 0.143) > ‘‘Road open space” (w = 0.088) > ‘‘Science education” (w = 0.074) > ‘‘Community ventilation” (w = 0.063). The results show that because of the limited scope of activities of the elderly after retirement, the scope of social interaction opportunities is further reduced. The community environment must first provide social functions for the elderly. This is to ensure that older people can interact with the community, improve their sense of belonging, and help maintain harmonious and stable mental health. The community environment should provide fitness and health functions for the elderly, which is important for ensuring the health of the elderly.

Among the six evaluation indicators of "landscape greening", the order of importance is “Landscape accessibility” (w = 0.268) > ‘‘Road landscape” (w = 0.259) > ‘‘Public landscape” (w = 0.204) > ‘‘Plant health” (w = 0.155) > “Ground landscape” (w = 0.082) > ‘‘Water sculpture” (w = 0.032). The results show that landscape environment accessibility is the most important assessment indicator of "landscape greening”. The beautification of the community environment ensures that the elderly can have barrier-free access to the landscape environment. 

## 6. Conclusions

In this research, a hierarchy of assessment indicators for community environment aging was built by pooling expert opinions and using the fuzzy analytic hierarchy process (FAHP) to obtain relative weights for appropriate aging indexes. On this basis, a comprehensive evaluation index system for community public environment aging was constructed. The study results can help academic researchers, practitioners, and industry decision-makers to find improvement solutions and evaluate these solutions.

### 6.1. Research Conclusions

In this study, through a professional questionnaire survey and factor analysis, a hierarchical structure of an assessment for the aging-friendly community public environment was obtained. It includes four factors and 24 detailed indicators, including ‘‘facilities”, “road system’’, “environmental function”, and ‘‘landscape greening”. These aspects and indicators are key factors that affect the evaluation of the community public environment. Through these detailed indicators, it is possible to carry out a more complete and objective assessment of the construction of the aging-friendly community public environment.In this study, the relative weight values of four factors and 24 project evaluation indicators of the aging-friendliness of the public environment of the community were obtained through the fuzzy analytic hierarchy process (FAHP). The results reveal that, whether it is a new community or an old community renovation, as long as it involves the aging of the community public environment, "facilities" and "road system" are the two core aspects of the construction situation to evaluate the aging-friendliness of the community public environment. The two aspects "environmental function" and "landscape greening" are more effective as supplements to the construction of the community public environment [1].Previous research on community aging has focused on problems and suggestions in the construction and management of the elderly care system and community living environment under the community home care model, and few materials are related to the evaluation system of the aging-friendly community public environment. In this study, through a review of national policy documents and standards, combined with a questionnaire survey of community-based elderly care-related personnel, rigorous research was conducted to obtain objective evaluation indicators and relative weights among indicators for future government departments to conduct community public environment construction. The results can provide reference guidelines during the planning stage.

### 6.2. Research Recommendations

In this research, an initial aging-friendliness evaluation index and its weights were constructed for the assessment of the community public environment. Future research directions will be based on verifying the results through the analysis of actual cases. For example, it is suggested to test whether the evaluation index weighting system constructed in this research can effectively evaluate the effectiveness of the construction of the aging-friendly public environment in a single community or multiple communities.This research mainly used experts in environmental planning and old-age security to quantitatively calculate the weight of the aging-friendly community public environment evaluation index. For future studies of this kind, elderly people who lack a local pension and the employees of community pension institutions could provide deeper insights into the weights of various indicators of the community public environment. In future extended research, we will further analyze the needs of the elderly for an aging-friendly environment, to supplement the shortcomings of the quantitative assessment from an expert perspective.This research mainly adopted domestic norms and standards and expert opinions. It is recommended to draw on developed countries’ normative standards and the opinions of experts of different nationalities on the evaluation of the community public environment’s fitness. This might not only allow more protracted research on the assessment indicators of the aging-friendly community public environment, but also help develop a global strategy for home care in the community. It can also further analyze differences in the views of domestic and foreign experts on this issue.At present, the research on age-appropriate construction in the community home-based old-age care model mainly focuses on existing problems and suggestions, regarding the construction and management of old-age care services, living environments, and social support. There is little research on how to effectively quantify the level of old-age-appropriate construction in a community. In particular, there is still a gap in the research on the evaluation system and index quantification of the aging-friendly community. In view of this, on the basis of the preliminary construction of the assessment system and weights for the aging-friendly community public environment, a follow-up discussion will focus on the humanistic care and social and economic aspects of the community, in the hopes of building a complete evaluation index system for the aging-friendly community under the model of community home care. It can be used to assess the needs of the existing community before the construction of an aging-friendly community environment, or provide a reference for the evaluation of the effects after its construction.

## Figures and Tables

**Figure 1 ijerph-17-02863-f001:**
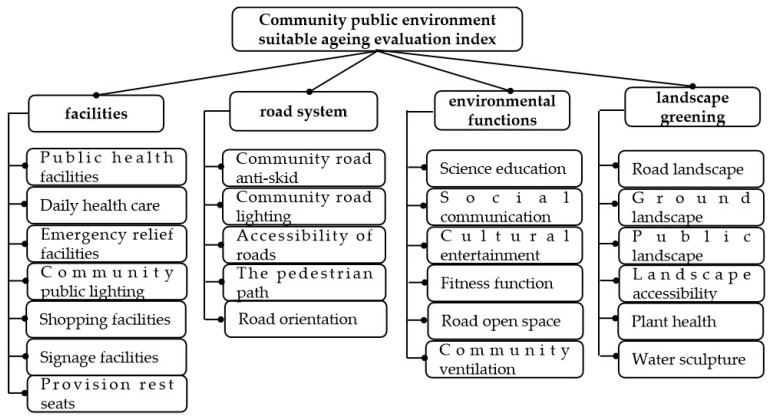
Aging-friendly community public environment suitable evaluation index.

**Table 1 ijerph-17-02863-t001:** Instruction Sheet for Filling Out the Fuzzy Hierarchy Analysis Expert Questionnaire.

Questionnaire Content	Purpose of the Survey
1. Completion instructions: Explain in detail how to fill out the questionnaire and use examples to explain	Use simple instructions to make it easier for respondents to fill out the questionnaire and save time.
2. Aging-friendly community public environment evaluation index framework: 4 factors, 24 indexes	Let the interviewees understand the structural relationship of each factor of the "Aging-Friendly Community Public Environment Evaluation Index System"
3. Fill in the questionnaire and explain the indicators: Respondents checked the “relative importance value”, “acceptable maximum value”, and “acceptable minimum value” by comparing the pairwise factors according to their importance (Table 2)	Ordering of factors: There are 4 factors: 1. facilities; 2. road system; 3. environmental functions; 4. landscape greening. If you think its order of importance is "1. facilities" > "3. environmental functions" > "2. road system" > "4. landscape greening", then please record (1) > (3) > (2) > (4). Relative importance of indicators: If you think that (i) the "relative importance" ratio of "public health facilities" to "Daily health care" is 5:1, (ii) the maximum acceptable value is 7:1, and (iii) the minimum acceptable value is 2:1, please separately tick (7:1), (5:1), (2:1) (Table 2).

**Table 2 ijerph-17-02863-t002:** Example of fuzzy level expert analysis expert questionnaire.

Impact Indicator	Strong Relative 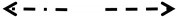 Importance Weak	Impact Indicator
9:1	8:1	7:1	6:1	5:1	4:1	3:1	2:1	1:1	1:2	1:3	1:4	1:5	1:6	1:7	1:8	1:9
**public health facilities**			√		√			√										Daily health care

**Table 3 ijerph-17-02863-t003:** Expert Basic Statistics.

Expert Member	Number of People	Proportion	Gender/Number	Education level/Number	Profession/Number
Professional research	4	33.3%	Male/3 Female/1	Bachelor/1 Master/1 Doctor/2	Industrial design/2 Environmental design/1 Social Security/1
Professional teaching	4	33.3%	Male/3 Female/1	Master/2 Doctor/2	Environmental planning/1 Environmental design/2 Industrial design/1
Professional practice	4	33.3%	Male/3 Female/1	Master/4	Environmental planning/2 Architectural design/2

**Table 4 ijerph-17-02863-t004:** Evaluation index preliminary selection statistics and interpretation table.

Evaluation Indicators	Indicator Definition Evaluation Content
Accessibility of roads	The slope of the road is reasonable in design and suitable in size, suitable for the elderly.
The pedestrian path	Diversion of pedestrians and vehicles in the community, with dedicated walkways
Community road lighting	Streetlights on both sides of community roads to ensure light and enable walking at night
Community road anti-skid	Community floor has a good non-slip property, suitable for elderly people
Road orientation	The road marking system in the community is continuous and highly directional
Road open space	The walking path in the community is not too long, and the step distance is reasonably controlled
Community lighting	Community interaction space has sufficient sunshine conditions
Community ventilation	Community communication spaces have adequate ventilation
Community noise	Community interaction space is quiet or noiseless
Cultural entertainment	Community interaction space can meet the cultural and entertainment functions of the elderly
Science education	Community interaction space can meet the popular science education function of the elderly
Fitness function	Community interaction space can meet the fitness and health functions of the elderly
Social communication	Community interaction space can provide the function of social interaction for the elderly
Public landscape	Community public environment is well greened and beautified
Road landscape	Planting landscapes on both sides of the road to improve walking comfort
Ground landscape	Beautiful pavement landscape to beautify the environment
Water sculpture	Community set-up of water sculpture landscape to beautify the environment
Landscape accessibility	Community landscape environment is barrier-free, suitable for elderly people
Plant health	Plants grown in the community are not harmful to human health
Provision rest seats	Community provides sitting chairs to meet sitting needs
Shopping facilities	Comprehensive shopping facilities and convenient daily life
Daily health care	Daily health and medical functions are provided to meet daily inspection medical and health consultation needs
Public health facilities	Complete public health facilities to provide convenience for daily life
Community public lighting	Community public lighting facilities are ideal to meet night lighting needs
Emergency relief facilities	Emergency relief assistance facilities are provided to meet emergency needs
Signage facilities	Signs allow easy identification in areas in which people are likely to get lost, improve recognition

**Table 5 ijerph-17-02863-t005:** The result of factor analysis extraction.

Evaluation Index	Factor 1	Factor 2	Factor 3	Factor 4
Public health facilities	0.850			
Daily health care	0.846			
Emergency relief facilities	0.795			
Community public lighting	0.780			
Shopping facilities	0.755			
Signage facilities	0.696			
Provision rest seats	0.615			
Community road anti-skid		0.806		
Community road lighting		0.806		
Accessibility of roads		0.763		
The pedestrian path		0.712		
Road orientation		0.572		
Science education			0.776	
Social communication			0.744	
Cultural entertainment			0.655	
Fitness function			0.650	
Road open space			0.640	
Community ventilation			0.594	
Road landscape				0.827
Ground landscape				0.792
Public landscape				0.764
Landscape accessibility				0.654
Plant health				0.605
Water sculpture				0.535
Eigenvalue	5.429	4.213	3.912	3.750
Square sum load extraction variation %	50.725	8.816	6.334	6.226
Shaft square and load variation %	22.621	17.555	16.301	15.624
Total variation (%)	72.101			
Cronbach’s α	0.957			
Total Cronbach’s α	0.931	0.890	0.899	0.905

**Table 6 ijerph-17-02863-t006:** Relative weight value of community public environment suitable aging evaluation index.

Facets	Weights	Sequence	Evaluation Index	Within Group		Whole Group		Inconsistency
				Weights	Sequence	Weights	Sequence	
**facilities**	**0.310**	**2**	**Public health facilities**	0.143	2	0.044	7	0.037
			Daily health care	0.381	1	0.118	2	
			Emergency relief facilities	0.142	3	0.044	8	
			Community public lighting	0.078	5	0.024	12	
			Shopping facilities	0.141	4	0.044	9	
			Signage facilities	0.044	7	0.014	19	
			Provision rest seats	0.071	6	0.022	14	
road system	0.374	1	Community road anti-skid	0.480	1	0.180	1	0.037
			Community road lighting	0.142	3	0.053	6	
			Accessibility of roads	0.220	2	0.082	4	
			The pedestrian path	0.113	4	0.042	10	
			Road orientation	0.045	5	0.017	16	
environmental functions	0.264	3	Science education	0.074	5	0.020	15	0.017
			Social communication	0.404	1	0.107	3	
			Cultural entertainment	0.143	3	0.038	11	
			Fitness function	0.228	2	0.060	5	
			Road open space	0.088	4	0.023	13	
			Community ventilation	0.063	6	0.017	17	
landscape greening	0.052	4	Road landscape	0.259	2	0.014	20	0.036
			Ground landscape	0.082	5	0.004	23	
			Public landscape	0.204	3	0.011	21	
			Landscape accessibility	0.268	1	0.014	18	
			Plant health	0.155	4	0.008	22	
			Water sculpture	0.032	6	0.002	24	

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
