# Peer review of "Research on Evaluation Indexes and Weights of the Aging-Friendly Community Public Environment under the Community Home-based Pension Model"

_ijerph, 2020, doi:10.3390/ijerph17082863_

Round 1
Reviewer 1 Report
The addressed topic is of utmost relevance and contemporaneity.
Nevertheless, it is not clear the concept of family pension and social pension functions. It’s central on the paper. Or it is not? What kind of relation with Chinese new pension model?
It is not also clear the relation between real estate industry and the functions of Chinese communities. What kind of transformations? What relation to real estate investment? What the relations to elders separation from “their children”… real children?
The literature review is about home care and not with the concept of pensions. What´s the relation to the paper? In a western point of view, they are both important but not the same discussion.
What is the thesis of the paper as pointed in line 190 and 215?
Page 289 and 290, must be explained, what kind of fitness indicators?
Page 316 What kind of evaluations items was collected. All the same nature, representing different issues? A list must be annexed and discussed, and how they were measured. What kind of scale was used? Interval or ratio? Or rank? That´s matter, because the type of analysis is different for each case.
Pag 331 Set 200 valid questionnaires for universe of how many analysis units?
Pag. 345 Was PCA extraction method used or other? (Page 403 was PCA, ok) It was ideal for the case. I doubt.
And all is written from line 404 to 410 it is not clear?
Please reconsider the method approach.
Author Response
Response to Reviewer 1 Comments
Thanks again for your time!
Point 1:The addressed topic is of utmost relevance and contemporaneity.
Response 1:Thank you for your comments and approval.
Point 2: Nevertheless, it is not clear the concept of family pension and social pension functions. It’s central on the paper. Or it is not? What kind of relation with Chinese new pension model?
Response 2: Family pension and social pension are the early pension modes in China, they are not the focus of this paper, The paper focuses on the model of community home care. In the model of community home care, family and social pension can improve the quality of life of the elderly. Please see page 2 of the revised paper
Point 3: It is not also clear the relation between real estate industry and the functions of Chinese communities. What kind of transformations? What relation to real estate investment? What the relations to elders separation from “their children”… real children?
Response 3: The development of residential buildings in China from the original multi-storey and small high-rise buildings to the high-rise and super high-rise buildings not only improves the living conditions of Chinese people, but also beautifies the community environment and sets up supporting facilities for the community. Please see page 2 of the revised paper. In the traditional Chinese family endowment mode, the elderly live with their children, and the children take care of the elderly.Please see page 2 of the revised paper
Point 4:The literature review is about home care and not with the concept of pensions. What´s the relation to the paper? In a western point of view, they are both important but not the same discussion.
Response 4: Community home care is a new model of Chinese old-age care developed on the basis of social pensions and family pensions. The paper focuses on the construction of community public environment under the model of community home care.
Point 5:What is the thesis of the paper as pointed in line 190 and 215?
Response 5: "care facilities"and "accessibility" are the thesis of the paper as pointed in line 190 and 215
Point 6:Page 289 and 290, must be explained, what kind of fitness indicators?
Response 6: Page 289 and 290mainly introduces the research method and process of the paper, and 26 items are selected, as shown in table 4 of the paper.
Point 7:Page 316 What kind of evaluations items was collected. All the same nature, representing different issues? A list must be annexed and discussed, and how they were measured. What kind of scale was used? Interval or ratio? Or rank? That´s matter, because the type of analysis is different for each case.
Response 7: Based on the semantic analysis of the home care policy and the keyword query in the literature, 26 evaluation indexes were collected from the coding results. The collected indicators are all indicators related to the construction of community public environment suitable for aging. See table 4 for the name and interpretation of the indicators.
Point 8:Page 331 Set 200 valid questionnaires for universe of how many analysis units?
Response 8: A total of 205 questionnaires were sent out and 202 valid questionnaires were received, which were all involved in the discussion of the paper.
Point 9:Page 345 Was PCA extraction method used or other? (Page 403 was PCA, ok) It was ideal for the case. I doubt.
Response 9: In the research and application of many fields, it is often necessary to make a large number of observations on multiple variables reflecting things and collect a large amount of data for the purpose of analyzing and finding the rules. Principal component analysis and factor analysis are commonly used research methods.
Point 10:And all is written from line 404 to 410 it is not clear?
Response 10: The eigenvalue of factor 1 was 5.429, the eigenvalue of factor 2 was 4.213, the eigenvalue of factor 3 was 3.912, and the eigenvalue of factor 4 was 3.750, which explained the variable variation amounts of 50.725%, 8.816%, 6.334% and 6.226%, respectively. The study obtained four factors in which the total variation of 72.101% could be explained. Please See page 12 and table 5 of the revised paper for details.
Point 11:Please reconsider the method approach.
Response 11: Factor analysis is a statistical method to assist in reducing the number of variables, and Principal component analysis (PCA) is a technique used to extract factor dimensions in factor analysis, which is commonly used in factor analysis.That the experts can allow this paper to use this method.
Reviewer 2 Report
The English is understandable but needs polishing to reach publication ideals.
This could be a really good article, but I would like to see a strong manner about validation now there is only a brief and superficial intent on it
Introduction section should be updated by adding the objective and motivation
Section 2.1 heading should be start with “Community”.
Authors should add the reference related to FAHP into the work.
Discuss the major finding in a separate paragraph/section with respect to the existing study.
Author Response
Point 1:The English is understandable but needs polishing to reach publication ideals.
Response 1: I will work hard to improve my English writing and will ask the MDPI professional translators to help me with the corrections to reach the publishing level before the final publication
Point 2:This could be a really good article, but I would like to see a strong manner about validation now there is only a brief and superficial intent on it.
Response 2: This study tries to establish an indicator system and calculate the weight value of indicators mainly by means of questionnaire survey and expert analysis, so as to evaluate the age-appropriate construction status of the community public environment, and provide safer facilities for the elderly.
Point 3:Introduction section should be updated by adding the objective and motivation
Response 3: Yes, I revised the introduction. Please see page 2 of the revised paper
Point 4: Section 2.1 heading should be start with “Community”.
Response 4: Yes, Section 2.1 heading start with “Community”.
Point 5:Authors should add the reference related to FAHP into the work.
Response 5: Yes, the literature on FAHP has been added. Please see page 10 of the revised paper
Point 6: Discuss the major finding in a separate paragraph/section with respect to the existing study.
Response 6: Yes, At present, the research on age-appropriate construction in the community home-based old-age care model mainly focuses on the existing problems and suggestions in the construction and management of old-age care services, living environment and social support. There is little research on how to effectively quantify the level of old age-appropriate construction in a community, In particular, there is still a gap in the research on the evaluation system and index quantification of the community suitable for aging. In view of this, based on the preliminary construction of the assessment system and weight of community public environment suitability for aging,The follow-up discussion will focus on the humanistic care and social and economic aspects of the community, hoping to build a complete evaluation index system of the community suitable for aging under the model of community home care.It is used to assess the needs of the existing community before the construction of suitable for aging, or to provide reference for the evaluation of the effect after the construction of suitable for aging. Please see page 17 of the revised paper
Round 2
Reviewer 1 Report
Dear authors, considering you review, I send you my second review in 8 points:
- Despite the efforts of the authors, the questions asked are not answered.
- Although changing the Chinese pension model is not the focus of the article, its discussion is sufficiently important as the authors themselves put it in the paper title and develop the topic in the introduction. Although changing the Chinese pension model is not the focus of the article, its discussion is sufficiently important as the authors themselves develop the topic in the introduction. This discussion is far from be clear, namely when we consider your answer 4.
- At same time, I understand the changes on real estate and the meaning for improving quality of households. I doubt about the option to build from the “original multi-storey and small high-rise buildings to the high-rise and super high-rise buildings” would be an acceptable way to elder people. How do they feel, what is their opinion? Is that so perfect? The perfections is not a author’s point of view?
- The thesis is not “care facilities” and “accessibility”. That is not a thesis. Your paper could be a product of a thesis, but no thesis is expressed in this article. Your paper aim is about indexes and its weights to evaluate environment conditions to implement a public policy.
- I understand the variables and indicators, but I still don’t understand is fitness.
- When I read table 4 I can’t understand the which measure scale of each one. Is crucial to the choice of statistical methods.
- I still don´t understand if 200 questionnaires are adequate or not for the study.
- No doubt about factor analysis. Is adequate to describe variability and show latent variables. In any case, Factor Analysis is ideal in an exploratory stage and PCA to a stage when all variables are cleared set and describes well our study universe. But, the measure scale, what was the nature of each variable must be clarified, and it isn´t. If they are not measured in a ratio or interval scale, AF is not suitable.
I hope my comments have been constructive.
Best regards
Author Response
Point 1: Despite the efforts of the authors, the questions asked are not answered.
Response 1: Thank you very much for your constructive comments,Your opinion has provided great help to the research of the paper
Point 2: Although changing the Chinese pension model is not the focus of the article, its discussion is sufficiently important as the authors themselves put it in the paper title and develop the topic in the introduction. Although changing the Chinese pension model is not the focus of the article, its discussion is sufficiently important as the authors themselves develop the topic in the introduction. This discussion is far from be clear, namely when we consider your answer 4.
Response 2: China's pension model has gone through three stages. In stage 1, before the 1990s, the number of people over 65 was relatively small and the number of children was large. In addition, the social economy was underdeveloped at that time, and the population movement was not frequent. Most of the elderly choose the model of family pension, living with their children, which is also in line with the traditional Chinese living habits. In stage 2, after 2000, due to the influence of modern factors such as urbanization, declining birth rate, and globalization, it is becoming increasingly common for the elderly to live separately from their children. Especially under the dual influence of the declining birth rate and prolonged average life expectancy. During this stage, China's traditional family pension function is continuously weakening. Thus, the model of institutional pension is becoming more and more popular. In stage 3, the supply of professional institutions for the aged increasingly highlights the shortage of facilities for the aged, the high cost of the aged and the lack of warmth in the aged care. With the economic development and the rise of the real estate industry in recent decades, the functions of Chinese communities have become increasingly comprehensive, and the model of community home-based pension has gradually emerged and become the most important care model in China.
The discussion about "community home-based pension" is in the second chapter, Section 2.1 “ community home care model's connotation".
Point 3:At same time, I understand the changes on real estate and the meaning for improving quality of households. I doubt about the option to build from the “original multi-storey and small high-rise buildings to the high-rise and super high-rise buildings” would be an acceptable way to elder people. How do they feel, what is their opinion? Is that so perfect? The perfections is not a author’s point of view?
Response 3: Before the commercialization of real estate, China implemented a system of "welfare housing", Namely, funds for building the housing of community dweller is first collected by community management unit. Then, the housing will be allocated to community dweller to live in with free charge after the housings have been built. Therefore, in order to save construction costs, residential buildings can only meet the basic residential functions. For example, the majority of residential buildings adopt mixed structure, the buildings below 9 floors do not have elevators, and it will not spend more money to build related living facilities and beautify the environment. After the housing system reformation in China in 2000, real estate began to be commercialized. In order to attract consumers to buy houses, developers began to pay attention to the overall function of residential buildings. Residential buildings from the original multi-stories and small high-rise to high-rise and super high-rise development. The vertical traffic elevators, barrier-free facilities, lighting in public areas, and public rest spaces have all been significantly improved. On the other hand, the land area saved by reasonable planning can add more supporting functions, for example, increasing the residents' activity center, outdoor fitness facilities, etc. With the economic development and the rise of the real estate industry in recent decades, the functions of Chinese communities have become increasingly comprehensive, and the model of community home-based pension has gradually emerged and become the most important care model in China.
Point 4: The thesis is not “care facilities” and “accessibility”. That is not a thesis. Your paper could be a product of a thesis, but no thesis is expressed in this article. Your paper aim is about indexes and its weights to evaluate environment conditions to implement a public policy.
Response 4: This paper is a preliminary study and is not part of a master's or doctoral dissertation. The purpose of this paper is to evaluate the aging index and weight of community public environment under the model of community home care. The public environment of the aging community discussed in this thesis includes the two important concepts of "care facilities" and "accessibility" in terms of space conditions. This view comes from the results of different references studies ( Please see references13-19).
Point 5: I understand the variables and indicators, but I still don’t understand is fitness.
Response 5: Firstly, based on the semantic analysis of the pension policy and the keyword query in the literature, as a result, a total of 26 evaluation items were collected for this study. Secondly, the expert team is invited to conduct the screening of the index fitness to preliminarily adjust and modify the inappropriate indicators. Finally, according to the preliminary evaluation index of experts, Likert 5 scale was designed to conduct questionnaire survey on the importance and performance for each index, where 1 is very unimportant (very dissatisfied); 2 is not important (dissatisfied); 3 is ordinary; 4 represents important (satisfied); 5 is very important (very satisfied).
Point 6: When I read table 4 I can’t understand the which measure scale of each one. Is crucial to the choice of statistical methods.
Response 6: Table 4 explains the meaning of all 26 indicators. The indicators of this study were evaluated by Likert 5 scale, where 1 is very unimportant (very dissatisfied); 2 is not important (dissatisfied); 3 is ordinary; 4 represents important (satisfied); 5 is very important (very satisfied).
Point 7: I still don´t understand if 200 questionnaires are adequate or not for the study.
Response 7: In this study, 202 valid questionnaires were used to run different statistical analyses. The main supportive evidences for the study to use factor analysis are described as followings:
(1)The number of 200 questionnaires has reached the minimum sample size for the study of human behavior. References:Arrindell, W. A., & van der Ende, J. (1985). An empirical-test of the utility of the observations to-variables ratio in factor and components analysis. Applied Psychological Measurement, 9(2), 165-178. MacCallum, R. C., Widaman, K. F., Zahang, S., & Hong, S. (1999). Sample size in factor analysis. Psychological Methods,4(1),84-99.
(2) The number of questionnaire samples using factor analysis should be more than 5 times the number of variables in questionnaire. References: Chen, K. Y. & Wang , Z. H. (2017). Statistical analysis practice: SPSS and AMOS, Taipei: Wu-Nan Book. [in Chinese, semantic translation]
Point 8: No doubt about factor analysis. Is adequate to describe variability and show latent variables. In any case, Factor Analysis is ideal in an exploratory stage and PCA to a stage when all variables are cleared set and describes well our study universe. But, the measure scale, what was the nature of each variable must be clarified, and it isn´t. If they are not measured in a ratio or interval scale, AF is not suitable.
Response 8: Yes, the study used a five-point Likert scale to measure variables and the scale is treated as the interval scale which is suitable for factor analysis. After running the factor analysis, the KMO value of the variables in this study was 0.933, and the significance of Bartlett's sphericity was 0.000 which indicates that factor analysis is suitable for this study.
|
KMO and Bartlett verification |
||
|
Kaiser-Meyer-Olkin Measure the adequacy of sampling |
.933 |
|
|
Bartlett Spherical verification |
chi-square |
4022.752 |
|
df |
276 |
|
|
sig |
.000 |
|

Round 3
Reviewer 1 Report
Dear authors, considering you review, I send you my third comment on your paper.
Firstly, I must congratulate you for the improvement and polish of this version of your paper
your paper. Second, thank you for answer 2 and 3, it was very clear.
Nevertheless, I must point three comments.
First, when we “Chinese communities have been increasingly perfected “or “the perfection of China’s Smart new pension model” is supported in author´s opinion or Jing Tiankui, 2015 opinion? Supported in what evaluation. It was supported in a benchmarking analysis or assessment analysis? This is a scientific paper, not an opinion newspaper statement. So, this kind of statement must be proved or avoid it in this sense.
Second, as I expected, you said that measure scale was a Likert 5 scale. The problem is that the Likert scale is not an interval scale. Is a hierarchy one. The difference between satisfied and very satisfied is not the same as ordinary and satisfied. I know that is common the use of this scale as interval one but is not correct. My recommendation is to mention this in your paper, in the text or as footnote. That is why I first said that the use of PCA wasn´t the best way to analysis the inquiry answers.
Third, the justification for the sample dimension, I prefer that you use the second option, “The number of questionnaire samples using factor analysis should be more than 5 times the number of variables in questionnaire”, is in a statistical point of view much more sound, because the validation of the sample significance depends of the number of questions and the variance of each one.
Please consider those comments.
Author Response
Response to Reviewer 1 Comments-3
Thank you very much for your constructive comments, which are of great help to the research of the paper.
Point 1: First,when we “Chinese communities have been increasingly perfected “or “the perfection of China’s Smart new pension model” is supported in author´s opinion or Jing Tiankui, 2015 opinion? Supported in what evaluation. It was supported in a benchmarking analysis or assessment analysis? This is a scientific paper, not an opinion newspaper statement. So, this kind of statement must be proved or avoid it in this sense.
Response 1: Yes, in this paper, the view of "Chinese communities have been increasingly perfected" is quoted from the 2015 paper of scholar jing tiankui. However, this view is widely recognized in Chinese academia and the pension industry.
Point 2: Second, as I expected, you said that measure scale was a Likert 5 scale. The problem is that the Likert scale is not an interval scale. Is a hierarchy one. The difference between satisfied and very satisfied is not the same as ordinary and satisfied. I know that is common the use of this scale as interval one but is not correct. My recommendation is to mention this in your paper, in the text or as footnote. That is why I first said that the use of PCA wasn´t the best way to analysis the inquiry answers.
Response 2: Thank you for your suggestion. I will add the description that the measurement scale is likert 5 in the paper, please refer to the revised paper for details
Point 3:Third, the justification for the sample dimension, I prefer that you use the second option, “The number of questionnaire samples using factor analysis should be more than 5 times the number of variables in questionnaire”, is in a statistical point of view much more sound, because the validation of the sample significance depends of the number of questions and the variance of each one.
Response 3: Thank you very much for your suggestion. I will use the second option for the reasons of sample size.See the revised paper for details
